# Superconductivity enhancement and particle-hole asymmetry: Interplay with electron attraction in doped Hubbard model

Zhi Xu[1], Hong-Chen Jiang[2*] and Yi-Fan Jiang[1†]

**1** School of Physical Science and Technology, ShanghaiTech University, Shanghai 201210, China
**2** Stanford Institute for Materials and Energy Sciences, SLAC National Accelerator Laboratory and Stanford University, Menlo Park, CA 94025, USA

⋆ hcjiang@stanford.edu , † jiangyf2@shanghaitech.edu.cn

## Abstract

The role of near-neighbor electron attraction $V$ in strongly correlated systems has been at the forefront of recent research of unconventional superconductivity. However, its implications in the doped Hubbard model on expansive systems remain predominantly unexplored. In this study, we employ the density-matrix renormalization group to examine its effect in the lightly doped $t$-$t'$-Hubbard model on six-leg square cylinders, where $t$ and $t'$ are the first and second neighbor electron hopping amplitudes. In the electron-doped regime ($t' > 0$), we find that attractive $V$ can substantially enhance superconducting correlations, driving the system into a pronounced superconducting phase when the attraction exceeds a modest value $V_c \approx 0.5t$. In contrast, in the hole-doped regime ($t' < 0$), while heightened superconducting correlations have also been observed in the charge stripe phase, the systems remain insulating with pronounced charge density wave order. Our results demonstrate the importance of the electron attraction in boosting superconductivity in broader doped Hubbard systems and highlight the asymmetry between the electron and hole-doped regimes.

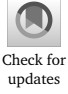

## Contents



# 1  Introduction

High-temperature superconductivity (SC) in cuprates is one of the central topics in the field of condensed matter physics [1, 2]. Despite substantial effort, understanding the microscopic mechanism underlying $d$-wave high-temperature SC remains highly challenging [3–35]. Among the various models proposed to understand SC in cuprates, the Hubbard model is one that has been extensively investigated. It is widely believed that this seemingly simple model can generate rich phases including antiferromagnetism, $d$-wave SC and various charge density waves and potential pair density waves [31–33, 35–42]. However, recent numerical studies suggest that the $d$-wave SC is absent in the pure Hubbard model with only nearest-neighbor (NN) electron hopping $t$ and strong on-site Coulomb repulsion $U$ [19, 20]. Therefore, it is natural to inquire about additional factors required beyond the simplest Hubbard model to realize $d$-wave SC.

With the significant advancements in numerical methods, many progresses have been made in searching potential factors that can induce or enhance SC by supplementing the Hubbard model. Recent density matrix renormalization group (DMRG) study on four-leg square cylinders has shown that the next-nearest-neighbor (NNN) electron hopping $t'$ can play an essential role in tipping the balance between SC and charge density wave (CDW) order such that quasi-long-range SC can be realized in the doped $t$-$t'$-Hubbard model [18, 19]. However, a more recent DMRG study on the wider six-leg square cylinders found that $d$-wave SC was only observed for the electron-doped case with positive $t'$ but not for the hole-doped case with negative $t'$ [25], while the DMRG study on closely related $t$-$J$ model [26, 43] and constraint path Quantum Monte Carlo study on $t$-$t'$-Hubbard model suggest that the SC may persist in hole-doped ($t' < 0$) region on wider systems [30]. On the other hand, experimental and theoretical studies on cuprates and Fe-based superconductors [44–51] have shed light on the electron-phonon coupling (EPC) as a potentially important ingredient in the Hubbard model. For instance, in one-dimensional cuprate chains, both photo-emission experiments and related numerical studies have provided evidence of a strong phonon-mediated NN attractive interaction [46, 48, 49]. This is further supported by a recent DMRG study that this NN attraction could notably enhance the SC correlations in the negative-$t'$ Hubbard model on four-leg square cylinders [27]. More numerical evidence of $d$-wave SC ordering was also reported in the simplest Hubbard model accompanied by moderate Su-Schrieffer-Heeger EPC mediated by the oxygen sites [52–54]. However, the overall effect of this electron attraction in the doped Hubbard model encompassing both electron- and hole-doped sides on wider systems still remains largely to be explored.

In this study, we have performed large-scale DMRG simulation to investigate the influence of NN electron attraction $V$ on the diverse phases exhibited by the $t$-$t'$-Hubbard model on six-leg square cylinders. Our results uncover a significant interplay between NN electron attraction $V$ and SC within the extended Hubbard model. Notably, NN electron attraction substantially bolsters superconductivity while concurrently mitigating CDW stripe order in the electron-doped ($t' > 0$) region of the model where we find that a moderate $V$ is sufficient to drive

the system into a pronounced SC phase with dominant SC correlations. Conversely, although NN attraction markedly intensifies SC correlations on the hole-doped ($t' < 0$) side of the phase diagram, our findings indicate that this electron attraction is insufficient to transition the systems into a SC phase. Consequently, despite the enhancement in SC correlations, the system remains still insulating in the hole-doped case.

## 2 Model and method

We employ DMRG method [55, 56] to study the ground state of the lightly doped extended Hubbard model on the square lattice defined by the Hamiltonian

$$H = -\sum_{ij\sigma} t_{ij} \left( \hat{c}^{\dagger}_{i\sigma} \hat{c}_{j\sigma} + h.c. \right) + U \sum_{i} \hat{n}_{i\uparrow} \hat{n}_{i\downarrow} - V \sum_{\langle ij \rangle} \hat{n}_i \hat{n}_j. \tag{1}$$

Here $\hat{c}^{\dagger}_{i,\sigma} (\hat{c}_{j\sigma})$ is the creation (annihilation) operator of spin $\sigma$ electron on site $i$. The electron hopping amplitude $t_{ij} = t$ for NN bond and $t'$ for NNN bond. $\hat{n}_{i\sigma} = \hat{c}^{\dagger}_{i\sigma} \hat{c}_{i\sigma}$ is the electron density operator of spin $\sigma$. The on-site Hubbard repulsion is denoted by $U$, and the NN attractive interaction is $V$. We take the lattice geometry to be cylindrical with periodic boundary condition in the $\hat{y} = (0, 1)$ direction and open boundary condition in the $\hat{x} = (1, 0)$ direction. We consider finite square cylinders with width $L_y$ and length $L_x$, where $L_x$ and $L_y$ are the number of sites along $\hat{x}$ and $\hat{y}$ directions, respectively. The doping concentration is defined as $\delta = N_h/N = 1 - N_e/N$, where $N_h = N - N_e$ is the number of doped holes, $N_e$ is the number of electrons in the system and $N = L_x \times L_y$ is the number of sites. For the present study, we focus on six-leg square cylinders with width $L_y = 6$ and various length $L_x$ and for simplicity fix the hole doping concentration $\delta = 1/12$. We set $t = 1$ as the unit of energy and report results for $V \leq 1.0$ and $U = 12$. We perform up to 130 sweeps and kept up to $m = 43,000$ states in each DMRG block to reach a typical truncation error of $\epsilon \lesssim 3 \times 10^{-6}$.

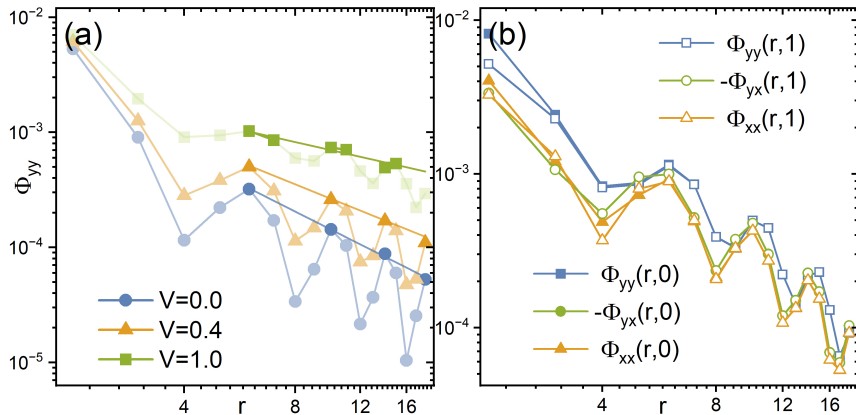

Figure 1: (a) SC correlation functions for $t' = 0.4$ and $U = 12$ on $N = 36 \times 6$ cylinder at $\delta = 1/12$ and different $V$. Solid lines denote power-law fit $\Phi(r) \propto r^{-K_{sc}}$ using the dark-colored points. (b) SC correlations between different types of bonds for $V = 1.0$ measured with $m = 40000$ kept states.

## 3 Enhanced SC correlation in $d$-wave phase

Previous DMRG studies of the doped $t$-$t'$-Hubbard model have observed the simultaneous presence of quasi-long-range $d$-wave SC and CDW orders in the electron-doped ($t' > 0$) region on both four-leg and six-leg square cylinders. The Luttinger exponents $K_{sc}$ and $K_{cdw}$, derived from the decay rate of the correlation functions for SC and CDW orders, were found to be comparable [18, 19, 25]. However, within a specific region of the phase diagram, the SC correlation was weaker than the CDW order, characterized by $K_{sc} > K_{cdw}$. This observation prompts an intriguing question: can the introduction of NN electron attraction strengthen the SC correlation enough to make it the predominant order in this coexistence phase?

To answer this question, we have calculated the equal-time spin-singlet SC correlation function defined as

$$\Phi_{\alpha\beta}(r) = \left\langle \Delta^\dagger_{\alpha,(x_0,y_0)} \Delta_{\beta,(x_0+r,y_0)} \right\rangle, \tag{2}$$

where $\hat{\Delta}^\dagger_{\alpha,(x,y)} = \frac{1}{\sqrt{2}}\left(c^\dagger_{\uparrow,(x,y)}c^\dagger_{\downarrow,(x,y)+\alpha} - c^\dagger_{\downarrow,(x,y)}c^\dagger_{\uparrow,(x,y)+\alpha}\right)$ creates a spin-singlet pair on bond $\alpha = \hat{x}$ or $\hat{y}$ emerged from site $(x, y)$. $(x_0, y_0)$ is the reference site with $x_0 \sim L_x/4$ and $r$ is the distance between two bonds along the $\hat{x}$ direction.

We first calculate the SC correlation functions $\Phi_{\alpha\beta}(r)$ for $t' = 0.4$ on six-leg square cylinders with $V = 0 \sim 1.0$ and an interval of $\Delta V = 0.2$. As shown in Fig.1(a), the amplitude of $\Phi_{\alpha\beta}(r)$ increases monotonically with $V$ and decays as a power law at long distances as indicated by the solid lines. To quantitatively understand how the electron attraction ($V$) impacts the long-distance decaying behavior of SC correlations, we have fitted the SC correlations use a power-law function $\Phi(r) \propto r^{-K_{sc}}$. Here, $K_{sc}$ represents the Luttinger exponent, a crucial parameter that describes the quasi-long-range nature of SC correlations. Starting with $V = 0.0$, our analysis revealed that the Luttinger exponent $K_{sc} = 1.5(2)$, which aligns well with results from earlier study [25]. Interestingly, as we incrementally increased $V$ from 0.0 to 1.0, we noticed a significant trend: the exponent $K_{sc}$ consistently decreased, reaching down to $K_{sc} = 1.0(1)$. This trend suggests that as temperature $T$ approaches 0, the SC susceptibility $\chi_{sc} \sim T^{-(2-K_{sc})}$ becomes increasingly divergent for stronger electron attraction. Our findings from the six-leg square cylinders align with those observed in the narrower four-leg square cylinders [27]. This consistency underscores the overarching role of electron attraction in amplifying SC correlations across different system widths.

The pairing symmetry of SC order is determined by measuring the SC correlation functions $\Phi_{\alpha\beta}(r, \delta_y) = \left\langle \Delta^\dagger_{\alpha,(x_0,y_0)} \Delta_{\beta,(x_0+r,y_0+\delta_y)} \right\rangle$ between the two bonds separated by $\delta_y$ sites along $\hat{y}$ distance. For instance, as depicted in Fig.1(b) for $V = 1.0$, we observe that the relationships $\Phi_{yy}(r, 0) \sim \Phi_{xx}(r, 0) \sim -\Phi_{yx}(r, 0) \sim \Phi_{yy}(r, 1)$ are maintained. This pattern indicates that the $d$-wave pairing symmetry, identified in earlier research [25], continues to be prevalent even when the electron attraction is strong, up to a value of $V = 1.0$.

**Charge density wave:** To identify the prevailing quasi-long-range order within the phase where SC and CDW orders coexist, we have also examined the behavior of long-distance correlations in the charge density channel. For instance, we calculate the charge density profile $n(x, y) = \langle \hat{n}(x, y)\rangle$ and its rung average density $n(x) = L_y^{-1}\sum n(x, y)$ at $\delta = 8.33\%$ with different $V$. The evolution of charge density as the attraction $V$ increases is detailed in Fig.2. The density profile $n(x)$, depicted in Fig.2(a), shows a pattern of clear spatial oscillations which have a wavelength of approximately $\lambda \approx 1/3\delta = 4$ in the $\hat{x}$ direction, indicative of a stripy CDW characterized by an ordering vector $Q \approx 6\pi\delta$. These oscillations, induced by the open boundary, manifest a power-law decay moving into the bulk, and can be fitted by the Friedel oscillation [57]

$$n(x) = A\cos(Qr + \phi)x^{-K_{cdw}/2} + n_0. \tag{3}$$



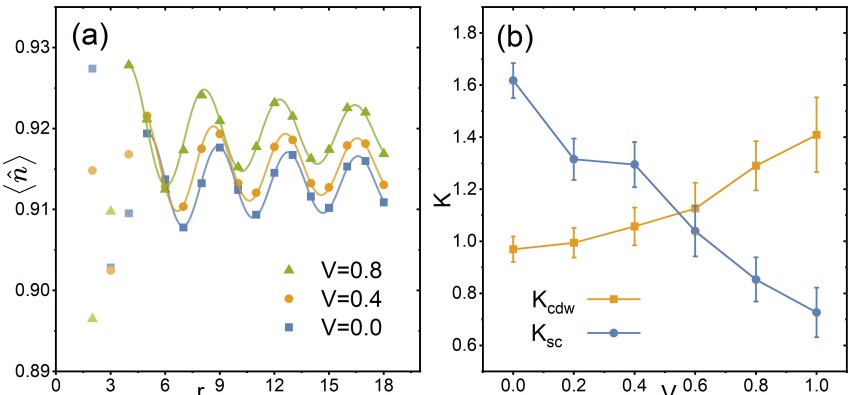

Figure 2: (a) Charge density profile $n(x)$ at $\delta = 1/12$, $t' = 0.4$ with different $V$ on $N = 36 \times 6$ cylinder. Solid lines denote the Friedel oscillation. Data points from the open boundary are omitted to avoid boundary effect. (b) Extracted exponents $K_{sc}$ and $K_{cdw}$ as a function of $V$ for the $\delta = 1/12$ models.

Here, $n_0$ is the average charge density, $K_{cdw}$ is the Luttinger exponent of the CDW order, $A$ and $\phi$ are model dependent parameters. When $V = 0$, the value of the exponent is $K_{cdw} = 0.8(4)$, which is found to be less than the exponent for the SC correlation, $K_{sc} = 1.5(2)$. The product of exponents $K_{sc}K_{cdw} \sim 1.2$ indicates a slightly deviation from the 1-D Luther-Emery liquid observed on 4-leg cylinders [19]. The relation $K_{sc} > K_{cdw}$ suggests that within a certain area of the phase diagram the CDW correlation might be dominant, agreed with the previous results of extended Hubbard model on 6-leg ladder [25].

Adding a small attractive $V$ slightly raises the average electron density in the center, as illustrated in Fig.2(a), as this attraction energetically prefers fewer electrons at the edges. However, this small change does not significantly alter the ordering wavevector of the CDW. When the attraction strength exceeds approximately $V \approx 0.4$, there's a noticeable increase in the exponent $K_{cdw}$, corresponding to a clear weakening of the CDW order. We present an overview of how the exponents for both the SC and CDW orders change with $V$ for $t' = 0.4$ in Fig.2(b). Here, a clear observation is that SC order starts to dominate when the attraction strength surpasses the intersection point, around $V_c \approx 0.5$, of the two trend lines. Interestingly, the critical value of $V_c$ needed to enhance SC order in six-leg square cylinders is noticeably lower than that for four-leg square cylinders [27]. This may indicate that electron attraction plays a more important role in promoting SC order in wider systems.

**Other properties:** The single particle properties of the system is examined by measuring the equal-time Green's function,

$$G(r) = \left\langle c^{\dagger}_{(x_0,y_0),\sigma} c_{(x_0+r,y_0),\sigma} \right\rangle. \tag{4}$$

Contrary to both the SC and CDW correlations, the single-particle Green function decays exponentially at long distances and can be well fitted by an exponentially decaying function $G(r) \sim e^{-r/\xi_G}$, where $\xi_G$ is the single particle correlation length. For the $t' = 0.4$ case, a long but finite correlation length, $\xi_G \approx 20$, is observed across a range of attraction strengths $V$ from 0.0 to 1.0 (Details are in the supplemental material (SM)). This indicates that the system's single particle gap remains small but largely unaffected by $V$. This observation aligns with previous DMRG findings for the model when $V = 0.0$ [25].
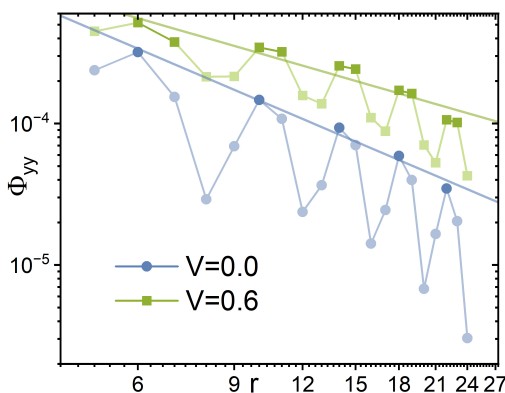

Figure 3: SC correlation functions measured on $N = 48 \times 6$ cylinder at $\delta = 1/12$ for $t' = 0.4$ and different $V$. Solid lines denote the power-law fits $\Phi(r) \propto r^{-K_{sc}}$ using dark-colored data points.

Due to constraints from the DMRG block dimension in our simulations on six-leg cylinders, the local spin $\langle S_i^z \rangle$ remain finite at each site. However, the profile of $\langle S_i^z \rangle$ exhibits a simple period-2 antiferromagnetic pattern, lacking any anti-phase domain walls and showing little to no dependence on the attraction $V$ strength (more details are provided in the SM).

**Results on longer cylinders:** To ensure the accuracy of our results over long distances, we have carefully examined the potential impact of finite size effects by analyzing the SC pair-pair correlation functions on longer cylinders. As shown in Fig.3, we increase the cylinder length from $L_x = 36$ to $48$ and measure the SC correlation $\Phi_{yy}(x)$ for systems with $V = 0.0$ and $0.6$. Consistent with the results for $L_x = 36$ shown in Fig.1, $\Phi_{yy}$ for $L_x = 48$ also exhibits similar power-law decay at long distances across all cases examined. Notably, as the attraction $V$ increases from 0 to 0.6, the extracted exponents $K_{sc}$ decreases from 1.6(1) to 1.1(1) consistent with our observation on $L_x = 36$ cylinder. This suggests that the observed enhancement in SC correlation is retained even in larger systems.

# 4 Effect of attraction in other phases

The observed enhancement in SC correlations on the electron-doped (positive $t'$) side of the $t$-$t'$-Hubbard model on six-leg square cylinders prompts a natural inquiry into how electron attraction $V$ affects other phases within the model. Previous DMRG study [25] on six-leg square cylinders has shown that distinct phases can be realized in another side of the $t$-$t'$-Hubbard model at $\delta \approx 1/12$ with $t' \leq 0$. These include a unidirectional charge stripe phase near $t' = 0$ and a Wigner crystal (WC) phase around $t' \approx -0.4$ disrupting translational symmetry in both $\hat{x}$ and $\hat{y}$ directions, but SC correlations are short-ranged. As illustrated in Fig. 4(a), although the presence of electron attraction $V$ can also boost SC correlations in the charge stripe phase which is similar with the electron doped side, this enhancement doesn't transition the system into a SC phase, evidenced by the exponential decaying SC correlations. Alternatively, fitting the SC correlation with a power-law model results in a considerable exponent value $K_{sc} \approx 8$, suggesting the SC susceptibility remains finite. In the WC phase ($t' = -0.4$), the impact of $V$ is minimal due to the absence of Cooper pairs, with SC correlations for the $V = 0.0$ to $V = 1.0$ models remains nearly identical, as seen in Fig. 4(b).

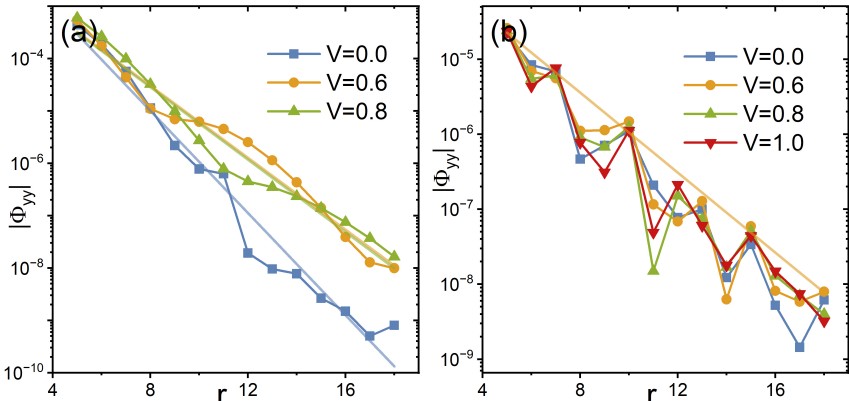

Figure 4: SC correlation functions $\Phi_{yy}(r)$ for (a) $t' = 0.0$ in the charge stripe phase and (b) $t' = -0.4$ in the holon Wigner Crystal phase. Color lines denote exponential fitting function $\Phi(r) \propto e^{-r/\xi_{sc}}$.

## 5 Summary and conclusion

In this study, we have explored the impact of NN electron attraction $V$ on SC and CDW orders within the $t$-$t'$-Hubbard model on six-leg square cylinders. We have shown that incrementing the attractive $V$ systematically amplifies SC correlations in both the SC and charge stripe phases. Notably, for electron-doped (positive $t'$) case, an increase in $V$ markedly enhances SC correlations while simultaneously diminishing CDW ordering, underscoring the pivotal role of electron attraction in augmenting superconductivity in the doped Hubbard model. The threshold attraction strength $V_c$, at which SC correlations become dominant over CDW correlations, is found to be around $V_c \approx 0.5$, in agreement with experimental observations on cuprates [46].

While we also note an increase in SC correlation in the hole doped side (negative $t'$) within the charge stripe phase, unlike the electron doped case, such an enhancement is insufficient to surpass the established long-range CDW order. Future research delving into mechanisms that could boost SC ordering in the negative $t$-$t'$-Hubbard model, such as the anti-ferromagnetic Heisenberg interaction induced by the Su-Schrieffer-Heeger type electron-phonon interaction [52–54] and the modulated hopping term mimicking the putative stripes [23], will be crucial. Delving into the synergy between these interactions could provide deeper insights into the mechanism of high-temperature superconductivity.

## Acknowledgments

We would like to thank Steven Kivelson, Thomas Devereaux and Dung-Hai Lee for insightful discussions.

**Funding information** Y.-F.J. acknowledges support from the National Program on Key Research Project under Grant No.2022YFA1402703 and from NSFC under Grant No.12347107 and 12574160. H.-C.J. was supported by the Department of Energy, Office of Science, Basic Energy Sciences, Materials Sciences and Engineering Division, under Contract DE-AC02-76SF00515.

# A   Supplemental material

## A.1   Finite truncation error extrapolation

We perform finite truncation error extrapolation to reduce the truncation error in our DMRG simulations with finite bond dimensions. The second-order polynomial $O(\epsilon) = O_0 + a_1 \epsilon + a_2 \epsilon^2$ is employed to extract the converged physical quantity $O_0$ in the zero truncation error limit $\epsilon \to 0$, based on the results $O(\epsilon)$ measured with $m = 25000$ to $43000$ block states. As illustrated in Fig. 5, we apply this extrapolation to both the pair-pair correlation function $\Phi(r)$ and the charge density $n(r)$ for the $t' = 0.4$ models with $V = 0.0$ and $0.8$ on $N = 36 \times 6$ cylinder. For the pair-pair correlation function $\Phi(r)$, extrapolations are independently applied to the data at each spatial separation $r$ to extract entire correlation functions in the zero truncation-error limit. In Fig. 5(a) and (b), we illustrate the curves of representative results for $\Phi(r = 14)$ in $V = 0.0$ and $0.8$ models. The fitting curves of rung average charge density $n(x)$ at site $x = 8$ are illustrated in Fig. 5(c) and (d). As local quantities, the charge densities exhibit weaker dependence on the truncation error, which can be well fitted even by linear functions.

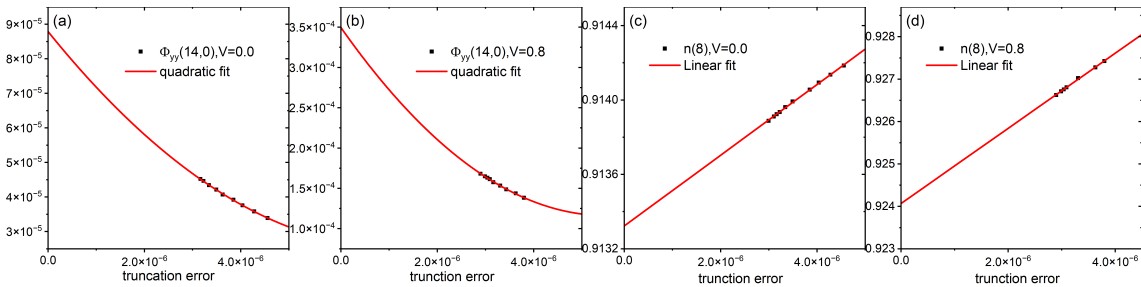

Figure 5: The examples of finite-truncation-error extrapolations for the pair-pair correlation function and rung average charge density: (a) the extrapolation of long-distance pair-pair correlation $\Phi(r = 14)$ for $V = 0.0$ model. (b) the extrapolation of $\Phi(r = 14)$ for $V = 0.8$ model. (c) the extrapolation of rung average charge density $n(x = 8)$ for $V = 0.0$ model and (d) $V = 0.8$ model.

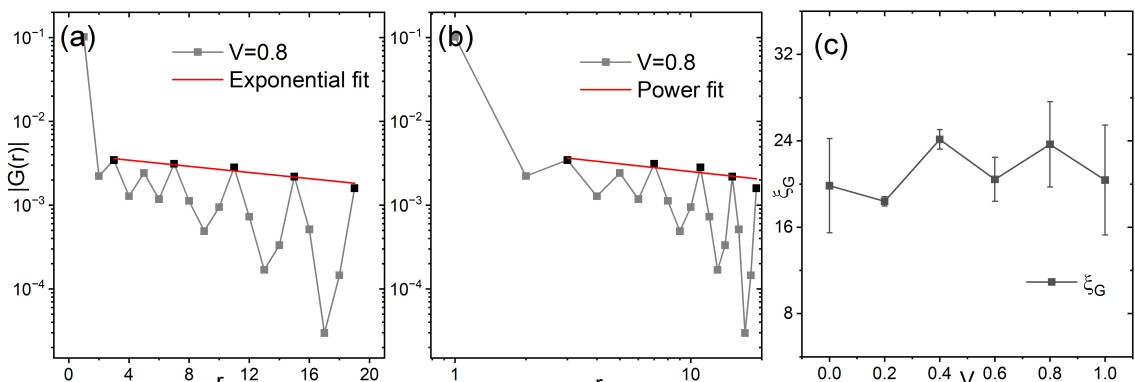

Figure 6: (a) The single particle correlation function $G(r)$ obtained from $t' = 0.4$, $U = 12$ and $V = 0.8$ model. Red line demonstrates the exponential fitting function of $G(r) \sim e^{-r/\xi_G}$ with correlation length $\xi_G \sim 23$. (b) The power-law fitting $G(r) \sim r^{-K_G}$ of the same data, with decay power $K_G \sim 0.3$. A clear deviation of the fitting curve appears at long distance. (c) The correlation length $\xi_G$ extracted from the exponential function for models with a range of attraction $V$ from $0.0$ to $1.0$.

### A.2 Single particle correlation function in $d$-wave SC phase

We measure the equal-time single particle correlation function $G(r)$ for the $t' = 0.4$ and $U = 12$ model with attraction $V$ ranged from 0.0 to 1.0. An example of the single particle correlation measured at $V = 0.8$ is illustrated in Fig. 6(a), exhibiting an exponentially decaying tail with a long but finite correlation length $\xi_G \sim 19$. The exponential decay of $G(r)$ is further supported by the power-law fitting shown in Fig. 6(b), where clear deviation from the fitting curve appears at long distances. The $V$-dependence of the correlation length $\xi_G(V)$ is summarized in Fig. 6(c), which suggests that the long-distance behavior of $G(r)$ remains largely unaffected by the attraction.

### A.3 Spin density profile in $d$-wave SC phase

Here we plot the profile of local spin momentum $(-1)^{x+y} \langle S_i^z \rangle$ for $t' = 0.4$ models with $V$ varied from 0.0 to 1.0, where the prefactor $(-1)^{x+y}$ explicitly encodes the period-2 antiferromagnetic background. All the results are measured with $m = 41000$ block states. As evidenced in Fig. 7, the spin profiles exhibit consistently simple period-2 antiferromagnetic pattern without any domain walls for all the attraction $V$ we tried.

### A.4 Full charge density profiles

We provide the full charge density profiles $\langle \hat{n}(x, y) \rangle$ for the models with different next-nearest-neighbor hoppings $t'$ in three phases. Figure 8 shows the charge density distribution for the $V = 0.6$ and $U = 12$ models on $N = 36 \times 6$ cylinders with (a) $t' = -0.4$, (b) $t' = 0.0$, and (c) $t' = 0.4$, providing a detailed view of charge pattern in different phases.

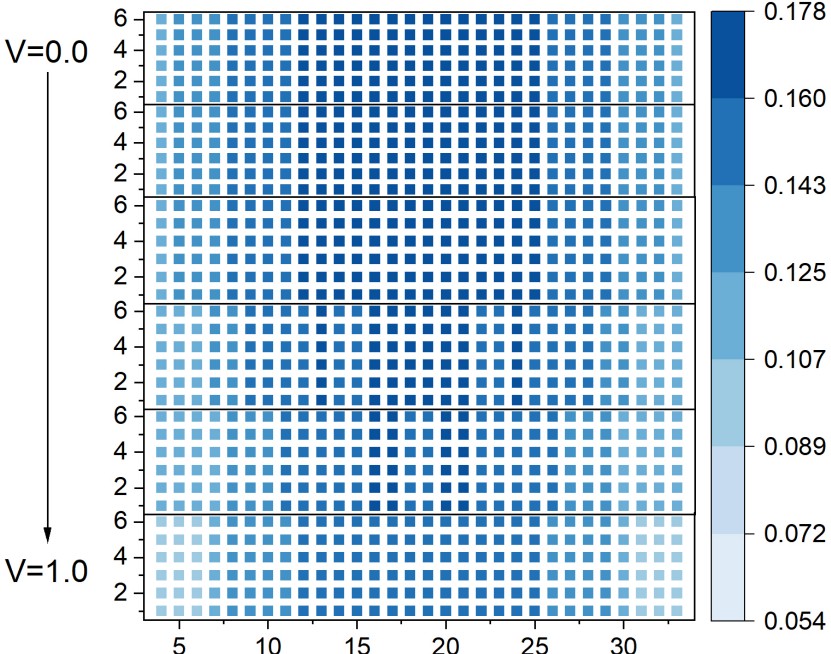

Figure 7: Spin density profile of the $t' = 0.4$ and $U = 12$ model with doping $\delta = 1/12$. The attraction $V$ increases from 0.0 (top) to 1.0 (down).

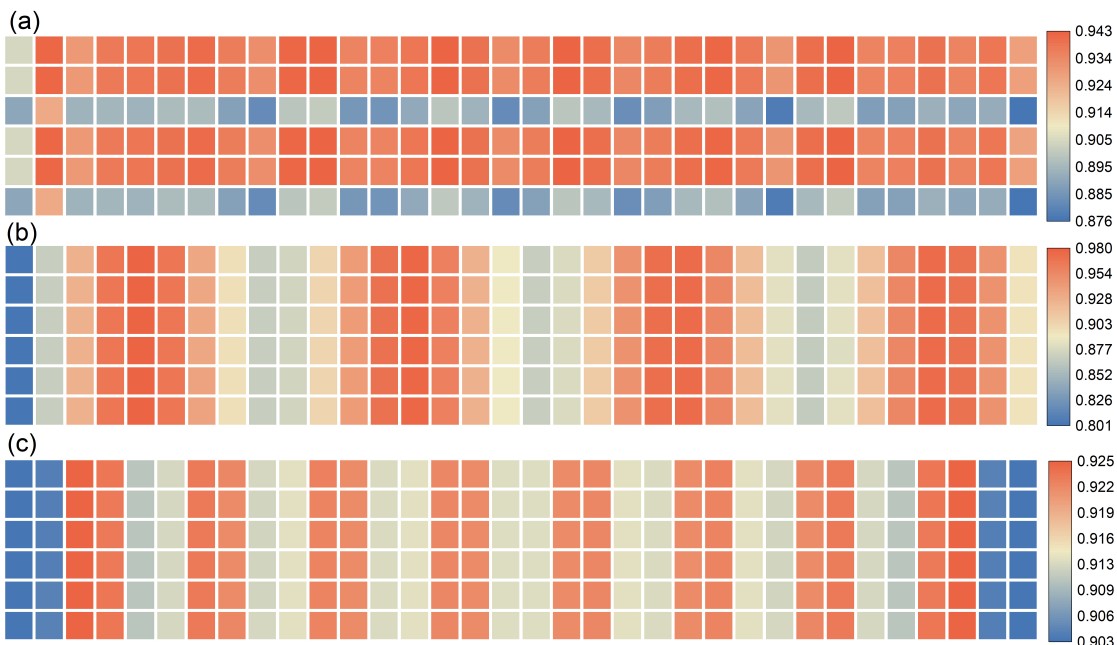

Figure 8: Full charge density profiles $\langle \hat{n}(x, y) \rangle$ of the $V = 0.6$ and $U = 12$ model with doping $\delta = 1/12$ and (a) $t' = -0.4$ in the WC* phase, (b) $t' = 0.0$ in the stripe phase and (c) $t' = 0.4$ in the SC phase.

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
