# Peer review of "Superconductivity enhancement and particle-hole asymmetry: interplay with electron attraction in doped Hubbard model"

_SciPost Physics, doi:SciPost Phys. 19, 137 (2025)_

## Round 1 · Referee Report · Anonymous (Referee 1) · 2025-9-30

Weaknesses

The power-law form of SC and CDW correlations, as well as the continuous change of the exponent as a function of interaction strength, is closely related to the case of the Luther-Emery liquid in one dimension. The numerical observation is very likely an artifact of DMRG calculations on long cylinders. It would be better if the authors could clarify this issue either by comparing the results for cylinders of different widths (e.g., 4-leg and 6-leg) or by providing theoretical references regarding the instability of coupled chains described by the Luther-Emery liquid fixed point.

Report

Using density matrix renormalization approach, the authors studied the doped t-t'-Hubbard model in the presence of a finite attractive nearest neighbour interaction. This calculation is very relevant for understanding the effect of electron phone effect for High-temperature superconductivity. The authors thoroughly investigated the competition between the d-wave superconductivity and the charge density wave order, and the enhancement of superconducting correlation is observed as the V interaction increases. On the other hand, the authors found no evidence of superconducting order in the case of hole doping. This work is timely, well written and definitely deserves publication.

Recommendation

Publish (meets expectations and criteria for this Journal)

---

## Round 1 · Referee Report · Anonymous (Referee 2) · 2025-10-5

Strengths

1- interesting finding that electron and hole-doped sides of the high-Tc Hubbard SC react very differently to the addition of a NN attractive interaction

Weaknesses

1- more data points would be needed on the hole-doped side (now only V=0 and V=0.6 t are shown) to make a truly convincing case.

Report

This paper by Xu et al. performs a highly interesting study, how the addition of a NN attraction to the t-t'-Hubbard model affects its superconducting properties. The study is extremely timely, the paper is well written and the results are notable: The authors find that the NN attraction V turns the system into a clean d-wave SC on the electron-doped side, whereas the SC correlations remain decaying exponentially on the hole doped side for the reported values of V. These results certainly fulfill the acceptance criteria of Sci Post Physics, if the following points can be addressed by the authors:

Requested changes

1- The authors should show more data for the hole-doped side, t'<0. Currently, only one set of results at V=0.6 t is shown, which (although impressive) in my view is not sufficient to make a really strong case. On the electron-doped side the authors scan V=0, 0.2 t, 0.4 t, ... 1.0 t in Fig. 2, and one keeps wondering why they didn't perform a similar analysis on the hole-doped side. I understand that in the Wigner crystal phase they cannot extract Luttinger exponents, but it would be interesting to show e.g. how the magnitude of the SC amplitude itself (at a certain distance, say) changes with V. It would also be interesting to see whether some response can be found before V exceeds 1.0 t, or if even larger values of V would be needed (eventually I assume the system should also become superconducting on the hole-doped side?).

2- it would be good to show some convergence data of the overall density profile. The oscillations in Fig. 2 appear well converged, but it is unclear what quantity exactly the authors plotted: is <n(x)> summed over all legs y? Can the authors directly show <n(x,y)> in the supplements, to give a better idea how well the data is converged?

Recommendation

Publish (surpasses expectations and criteria for this Journal; among top 10%)

---

## Round 2 · Author Response

Dear Editors,

Thank you for handling our manuscript "Superconductivity enhancement and particle-hole asymmetry: Interplay with electron attraction in doped Hubbard model" (scipost_202508_00079v1) and for the constructive feedback from the referees. We have carefully addressed all points raised in the reports through revisions and additional calculations. We believe these revisions have significantly strengthened our manuscript and thank the referees for their time and expertise.

Sincerely yours, Zhi Xu, Hong-Chen Jiang, and Yi-Fan Jiang

—————————————————— Responses to Report of Referee 1 —————————————————— Referee 1: Using density matrix renormalization approach, the authors studied the doped t-t′-Hubbard model in the presence of a finite attractive nearest neighbour interaction. This calculation is very relevant for understanding the effect of electron phone effect for High-temperature superconductivity. The authors thoroughly investigated the competition between the d-wave superconductivity and the charge density wave order, and the enhancement of superconducting correlation is observed as the V interaction increases. On the other hand, the authors found no evidence of superconducting order in the case of hole doping. This work is timely, well written and definitely deserves publication.

Reply: We thank Referee 1 for reviewing our paper and recommending it for publication.

  1. The power-law form of SC and CDW correlations, as well as the continuous change of the exponent as a function of interaction strength, is closely related to the case of the Luther-Emery liquid in one dimension. The numerical observation is very likely an artifact of DMRG calculations on long cylinders. It would be better if the authors could clarify this issue either by comparing the results for cylinders of different widths (e.g., 4-leg and 6-leg) or by providing theoretical references regarding the instability of coupled chains described by the Luther-Emery liquid fixed point.

Reply: We thank Referee 1 for the constructive suggestion. In the previous studies on narrow cylinders (e.g., refs[18,19,27]), it has been shown that the relationship $K_{sc} K_{cdw} = 1$ characterizing the 1-D Luther-Emery liquid holds in SC phases in both electron- and hole-doped side of phase diagram. While on the wider systems such as 6-leg cylinders, such relationship is no longer held in the SC phase. In our study of the models with NN interaction on 6-leg cylinders, we also observe that $K_{sc} K_{cdw} \ne 1$ in most of the cases. The deviation from the results on 4-leg cylinders highlights that the reduced quasi-1D character in wider systems. In revised manuscript, we have clarified this distinction in the section "enhanced SC correlation in d-wave phase" of the main text.

—————————————————— Responses to Report of Referee 2 —————————————————— Referee 2: This paper by Xu et al. performs a highly interesting study, how the addition of a NN attraction to the t-t′-Hubbard model affects its superconducting properties. The study is extremely timely, the paper is well written and the results are notable: The authors find that the NN attraction V turns the system into a clean d-wave SC on the electron-doped side, whereas the SC correlations remain decaying exponentially on the hole doped side for the reported values of V. These results certainly fulfill the acceptance criteria of Sci Post Physics, if the following points can be addressed by the authors:

Reply: We thank Referee 2 for the positive assessment of our work.

1- The authors should show more data for the hole-doped side, t′ < 0. Currently, only one set of results at V = 0.6t is shown, which (although impressive) in my view is not sufficient to make a really strong case. On the electron-doped side the authors scan V = 0, 0.2t, 0.4t, ...1.0t in Fig.2, and one keeps wondering why they didn’t perform a similar analysis on the hole-doped side. I understand that in the Wigner crystal phase they cannot extract Luttinger exponents, but it would be interesting to show e.g. how the magnitude of the SC amplitude itself (at a certain distance, say) changes with V . It would also be interesting to see whether some response can be found before V exceeds 1.0t, or if even larger values of V would be needed (eventually I assume the system should also become superconducting on the hole-doped side?).

Reply: We thank Referee 2 for the insightful question. To address the request regarding SC properties on the hole-doped side, we performed additional calculations for the models with t′ = −0.0 (stripe phase) and −0.4 (Wigner crystal phase) and updated results with more values of V in the Fig. 4 of the revised manuscript. As shown in the updated Fig. 4, the SC correlation of the t′ = −0.4 model remains short-ranged for V from 0 to 1.0. Moreover, the amplitude of SC correlations at long distance shows negligible dependence on the increasing V. Similar behavior of SC correlation is also observed in the t′ = 0 model for the stripe phase. For the larger value of V, we agree that other competing phases such superconductivity could appear if the phase coherence of paired electrons developed. We will explore the physics of models with extreme V in future study.

2- it would be good to show some convergence data of the overall density profile. The oscillations in Fig. 2 appear well converged, but it is unclear what quantity exactly the authors plotted: is ⟨n(x)⟩ summed over all legs y? Can the authors directly show ⟨n(x, y)⟩ in the supplements, to give a better idea how well the data is converged?

Reply: Yes, we sum over all legs for ⟨n(x)⟩. Following Referee 2’s suggestion, we add a new Fig. S4 in the supplemental material of the revised manuscript to directly show the full charge density profile ⟨n(x, y)⟩. The data confirm that the translational symmetry along the y-direction is well preserved.

---

## Round 2 · List of Changes

1. We have updated Fig. 4 to include additional SC correlation data obtained for models with different V.
  2. We have added one sentence to the section "Enhanced SC correlation in d-wave phase" of the main text to further clarify differences between results from 4-leg and 6-leg cylinders.
  3. We have added a new section "Full charge density profiles" in supplemental material.
  4. We have double-checked and updated references.

---

## Editorial Decision

published